# Nutritional Quality of Food and Beverages Offered in Supermarkets of Lima According to the Peruvian Law of Healthy Eating

**DOI:** 10.3390/nu12051508

**Published:** 2020-05-22

**Authors:** Mayra Meza-Hernández, David Villarreal-Zegarra, Lorena Saavedra-Garcia

**Affiliations:** CRONICAS Center of Excellence in Chronic Diseases, Universidad Peruana Cayetano Heredia, Lima 15074, Peru; mayrameza24@gmail.com (M.M.-H.); davidvillarreal@ipops.pe (D.V.-Z.)

**Keywords:** food and beverages, food composition, table of food composition, food labeling, food legislation

## Abstract

The purpose of this paper was to determine the foods and beverages offered in the city of Lima, Peru, that would be subject to front-of-package warning labels (octagons) according to the thresholds for the two phases (6 and 39 months after the approval) for nutrients of concern (sugar, sodium, saturated fat, and trans-fat) included in the Peruvian Law of Healthy Eating. An observational, descriptive cross-sectional study was conducted that evaluated the nutritional composition of processed and ultra-processed foods that are sold in a supermarket chain in Lima. Of all the processed and ultra-processed foods captured, foods that report nutritional information and do not require reconstitution to be consumed were included. A descriptive analysis was carried out by food categories to report the nutrient content and the percentage of foods that would be subject to front-of-package warning labels. *Results:* A total of 1234 foods were evaluated, according to the initial thresholds that became effective 6 months after the law was implemented; 35.9% of foods had two octagons; 34.8% had one octagon; 15.8% had no octagons; 12% had three octagons; and no products had four octagons. At 39 months, when the final and more restrictive thresholds become effective, 4.8% did not have octagons. The majority of processed and ultra-processed foods that are sold in a Peruvian supermarket chain carry at least one octagon, and more than 10% of them carry octagons for three of the four nutrients of concern.

## 1. Introduction

Currently, overweight and obesity continues to increase in children and adolescents [1]. This situation leads to the appearance, at an early age, of non-communicable diseases (NCDs) such as cardiovascular diseases, diabetes, some types of cancer, and others [2]. The rapid and constant increase in the consumption of processed and ultra-processed foods results in eating patterns that are based on foods containing higher energy density and high concentrations of fats, sugars, and sodium. Excessive consumption of these nutrients is one of the leading causes of obesity and the appearance of NCDs [3]. In Peru, the situation is not different. It has been reported that the largest growth in Latin America of retail sales of ultra-processed foods occurred in Peru [4].

In response to the evident relationship between the consumption of these foods with obesity and NCDs, countries in Europe [5], Oceania [6] and Latin America, such as Uruguay [7], Chile [8], and Ecuador [9] have implemented regulatory policies for advertising and labeling as well as establishing thresholds for energy and nutrients of concern for processed and ultra-processed foods. Some countries have implemented requirements for front-of-package warning labels for foods with nutrient content that exceeds nutritional thresholds established by their government. Requiring front-of-package warning labels is one of the measures that has the greatest impact on the population [10,11], consequently generating changes in their food preferences [12].

Like the countries mentioned above, in 2013, Peru passed the Law to Promote Healthy Eating for Children and Adolescents. This law mandated front-of-package warning labels, in the form of octagons, for processed and ultra-processed products. The law established thresholds for the content of sugar, saturated fat, sodium, and trans-fat. These thresholds determine which products must carry the front-of-package labels. Specifically, the thresholds define the upper limit for the amount of critical nutrients, per 100 g/mL of the product, that is allowed in order to avoid the requirement to carry the front-of-package warning label. The threshold values decreased, becoming more restrictive, with the final implementation of the law after 39 months.

Packaged foods that exceed the thresholds will carry the front-of-package warning label with either the phrase “high in” followed by the corresponding nutrient, either saturated fat, sugar, or sodium or the phrase “contains trans-fat” for foods that contain trans-fat. Therefore, a packaged food can carry up to four warning labels. The thresholds established by the law are to be implemented in two phases such that thresholds included in the second phase are more restrictive than those of the first phase. The first phase became effective 6 months after the approval of regulations based on the law. The second phase will be implemented 39 months after that approval. The thresholds and effective timeframes are detailed in Table 1.

The regulation governing trans fats became effective 18 months after it was approved. The regulation allows for the presence of trans fats in processed or ultra-processed foods and non-alcoholic beverages based on thresholds established in the regulation. The established thresholds are based on the type of food. First, fats, vegetable oils, and margarines are allowed up to 2 g of trans fats for every 100 g or mL of fat. Second, other processed products are allowed up to 5 g of trans fats for every 100 g or mL of food. All trans fats must be eliminated from processed or ultra-processed foods and non-alcoholic beverages after 54 months from the approval of the regulation.

Along with front-of-package warning labels and advertising regulations for processed and ultra-processed foods, the law also addresses the promotion of healthy eating and physical activity for children and adolescents in order to reduce NCDs [13]. The formulation of this law was based on policies that were implemented in and have been accepted by populations of other countries. For example, declaring the high content of a nutrient [14] and front-of-package warning labels [15] have been shown to be effective. Additionally, populations have given high approval to front-of-package warning labels. In some cases, similar policies have resulted in changes in food preferences [16]. They have also proven useful as a guide for consumers’ purchasing decisions [17]. The application of the regulation has also led to the reformulation of processed and ultra-processed foods, thereby reducing the content of critical nutrients in their composition [18,19] and as such contributing to improving their nutritional quality [20,21].

The present study seeks to determine which processed and ultra-processed foods offered in the city of Lima, Peru, would carry front-of-package warning labels according to the thresholds of the Peruvian law. Determining which products would carry front-of-package labels was based on the nutrient composition of products as offered prior to the implementation of the law. In addition, this study seeks to determine the amounts of critical nutrients in the products. This data will be useful to monitor the effects of the law. This data will also be useful in studies intended to assess the nutrient intake of children and adolescents.

## 2. Materials and Methods 

### 2.1. Study Design

This was an observational, descriptive, cross-sectional study. Data was collected at a single point in time during January and February 2018. The data collected at that time was used to determine which foods and beverages would carry a front-of-package label once the law went into effect in September 2018. Each product was assessed based on comparison of the amounts of critical nutrients to the respective thresholds using both sets of thresholds: those implemented in the first phase (at 6 months) and those to be implemented in the second phase (at 39 months).

### 2.2. Setting

Nutritional information for processed and ultra-processed food was collected from packaged products offered in three different supermarkets, each located in Lima, Peru. Each supermarket is part of a supermarket chain that, in turn, is a subsidiary of a larger retail company that has a national presence. Each of the three supermarket chains targets one of three particular socioeconomic sectors (high, medium, or low). 

### 2.3. Variables

The variables of interest were the proportion of processed and ultra-processed products that will carry front-of-package labels and also the amount of critical nutrients (that is, saturated fat, sugar, sodium, trans-fat) and the total energy in processed foods and beverages sold in Lima.

### 2.4. Data Sources

Nutrition information from labels on processed and ultra-processed food packages was the primary data source. Nutrition information was collected from photographs taken during January and February 2018. Photographs were taken, using cell phones, by six previously trained volunteer nutrition students at each of the three supermarkets mentioned above. The three supermarkets were those that are subsidiaries of the one larger retail company that granted access after requests were sent to several large retail companies that have supermarket chains in Lima, Peru. 

### 2.5. Procedures

Product packages were photographed as they appeared on the supermarket shelves from left to right, then top to bottom. A set of photographs was taken for each product. Each set contained a photo for each side of the package (i.e., top, bottom, front, back, right, and left). For cans, all photos necessary to capture the entire container were taken. Photographs of the barcode, list of ingredients and nutrition information were also taken (see Appendix A). For products available in different sizes, photographs were taken only for the size that occupied the majority of the shelf given that the nutritional composition was expected to be the same for all sizes.

Information on food labels, as seen in the photographs, was entered and stored in the web application, Food Nutrition Information Labeling Program (FLIP: Food Label Information Program), an application developed by the University of Toronto [22]. Data recorded in the system included net weight, servings per container, list of ingredients, nutritional information (energy, total fat, saturated fat, trans-fat, total carbohydrates, total sugars, and sodium) per portion or per 100 g or 100 mL. Later, the nutritional information for all products was standardized per 100 g or per 100 mL depending on the declared unit of measure. Finally, the products were grouped into one of 16 food categories. The categories originated with a model of categories used in a Chilean study [18] and adapted to the Peruvian context (see Appendix A).

### 2.6. Bias

The intent of the study was to capture and assess all processed products available in the market before the implementation of the front-of-package warning labels. To that end, products were photographed in three supermarkets known to serve three specific yet different socioeconomic sectors, thereby including products targeted at those varying sectors.

Data was validated in order to ensure that possible manual data entry errors were identified and corrected or excluded. Specifically, the Atwater System calculation was used to corroborate total energy from the package label with the sum of energy provided by each macronutrient using Atwater’s constants [23] (i.e., fats = 9 kcal/g, proteins = 4 kcal/g, carbohydrates = 4 kcal/g). This validation was applied to all products that declared each of the three macronutrients and energy. Foods with total energy values that equaled or were within 20% of the Atwater calculation for energy were included. Total sugar was compared to total carbohydrates for each product that provided both values. Products with total sugar greater than total carbohydrates were excluded from the analysis. Likewise, products with an amount of saturated fat that exceeded the amount of total fat were also excluded from analysis.

Dairy drinks were excluded from the analysis of trans-fat content because a determination could not be made as to whether the trans-fat amount declared in the nutritional information table were added or intrinsic in dairy products.

### 2.7. Sample Size and Data Characteristics

The sample size was based on the products available in each of the three supermarket chains at the time the photographs were taken during January and February 2018. Thus, the decision was made to use a non-probabilistic sample of convenience to assess all foods available at that time. Only one supermarket per supermarket chain was included because the supply of products between the supermarkets within a chain was very similar due to the fact that they have the same product providers. However, the food offered by different supermarket chains varies as each chain targets a different socioeconomic sector as noted above. The three supermarkets were selected to include products representative of those typically offered in each sector. 

All foods and beverages that were available on the date of the visit were included and evaluated since the objective was to capture the total supply of processed and ultra-processed foods. Minimally processed foods and culinary ingredients were also captured. However, three categories, fruits and vegetables, legumes, and oils were excluded from this study because the Peruvian law excludes foods in these categories from those that must carry front-of-package warning labels [24].

Food packages that did not provide nutritional information, or at least an energy amount (n = 657); and minimally processed food and ingredients (n = 569); and products that did not comply with the Atwater System validation (n = 62) were also excluded. In addition, foods that require reconstitution or preparation before consumption were also excluded (n = 225). According to the front-of-package warning label regulations, these foods must be assessed for front-of-package labels using nutrient values for the food as consumed not as purchased. Many of these foods could not be prepared because reconstitution or preparation instructions were not provided, exact amounts of added ingredients were not stated on the package, the weight or portion size of the prepared product was not available, or the composition of the added ingredient was not available (see Figure 1). 

### 2.8. Analysis of Data

First, the total number of foods and beverages captured was evaluated. Criteria described above were used to exclude products and determine those that would be included in further analysis. The criteria used included the Atwater System validation [15], comparison of total sugar and total carbohydrates, comparison of saturated fat and total fat, reconstitution requirements, and the existence of nutritional information. In addition, foods that could not be grouped into one of the 13 categories were also excluded. 

Second, for all remaining products, a descriptive analysis of frequencies, means, and standard deviations was performed for each of the 13 categories. The total number of foods and beverages was reported, as well as those reporting energy (kcal), saturated fats (g), sugar (g), sodium (mg), and trans-fat (g). Foods or beverages missing a particular nutritional value were not considered for analysis of that particular nutritional information. For example, a food or beverage without sodium on the package label was not included in the analysis of sodium. 

Finally, the number of foods that should carry one or more octagons was identified. The process of identification included comparing the value of each critical nutrient to the respective threshold value as defined in the Peruvian Law to Promote Healthy Eating for Children and Adolescents [8]. The comparison was made using both defined thresholds for a given critical nutrient. This allowed for two analyses to be performed, one analysis using the thresholds effective 6 months after implementation of the Law and a second analysis using the more restrictive thresholds to become effective 39 months after implementation of the Law. The ingredient list was used to identify foods that contains added trans fats. These analyses identified the number of foods that should have zero, one, two, three, or four octagons.

### 2.9. Ethics

The ethics committee of the Universidad Peruana Cayetano Heredia approved this study. This study is based on the information declared on labels on packages of processed and ultra-processed foods and, therefore, does not represent a risk for humans.

## 3. Results

### 3.1. Measurement Characteristics

Initially, 2747 foods were captured in the photographs taken. Of those, 2090 (76.1%) declared nutritional information and of those, 1521 (55.3%) were categorized as processed and ultra-processed foods. Subsequently, 1459 (53.1%) foods remained after applying both the Atwater and the macronutrient consistency criteria. Of those, 1234 (44.9%) remained that did not require reconstitution for consumption (see Figure 1). Finally, of the remaining 1234, 100% declared energy content, 87.8% declared sodium content, 87.8% declared total sugar content, and 74.6% declared saturated fat content.

### 3.2. Critical Nutrients by Food Category

Table 2 shows the amounts of nutrients per 100 g of food for each of the 13 food categories evaluated. The foods that had the highest concentration on average for each nutrient were candies (M = 12.9 g, SD = 8.6 g) and bakery products (M = 9.0 g, SD = 5.3 g) for saturated fats; candies (M = 53.6 g, SD = 16.5 g) and sweet spreads (M = 4.2 g, SD = 21.1 g) for sugar; condiments and spices (M = 9592.4 mg, SD = 11,138.2 mg) and sauces, spreads, and dressings (M = 931.2 mg, SD = 853.7 mg) for sodium; candies, which includes chocolates, (M = 0.3 g, SD = 1.1 g), and sauces, spreads, and dressings (M = 0.2 g, SD = 0.9 g) for trans-fat.

### 3.3. Food with Front-of-Package Warning Labels (Octagons)

Table 3 identifies the number of foods that did not require reconstitution for consumption that would carry a front-of-package warning label 6 months after implementation of the Law. Table 4 identifies the foods that would carry a front-of-package warning label 39 months after implementation of the Law.

The highest percentage of foods that required the front-of-package warning label for high saturated fat content based on the initial and less restrictive (higher) thresholds effective 6 months after implementation of the law were bakery products (73.4%) and candies (72.6%). These percentages increased to 82.7% and 74.9%, respectively, based on the final and more restrictive (lower) thresholds effective at 39 months. The percentage increases were due to the fact that the thresholds defining the allowed amount of critical nutrients per 100 g/mL of the product decreased and became more restrictive with the final implementation at 39 months. 

The highest percentage of foods that required the front-of-package warning label for high sugar content based on the initial and less restrictive (higher) thresholds were candies (95.5%) and sweet spreads (81.8%). The percentage for candies increased to 97.5% based on the final thresholds and for the same reason as stated above. The percentage for cereals increased to 90.2% displacing sweet spreads as the food category with second highest percentage.

The highest percentage of foods that required the front-of-package warning label for high sodium content based on the initial thresholds were processed condiments and spices (85.3%) and sauces, spreads, and dressings (50.0%). These percentages increased to 94.1% and 58.6%, respectively, based on the final thresholds.

The highest percentage of foods that required the front-of-package warning label for trans-fat content based on the initial thresholds were processed desserts (83.3%) and candies (21.3%). These percentages remained the same with the final threshold, since both the initial and the final thresholds were the same, namely any amount of trans-fat. 

Finally, 84.2% of food products should have had at least one front-of-package warning label based on the initial less restrictive thresholds. This percentage increased to 95.2% based on the final more restrictive thresholds. The percentage of foods that would carry four, three, two, or one front-of-package warning label octagon can be seen in Figure 2.

## 4. Discussion

One of the findings of this study shows that the categories with foods that carry the most front-of-package warning labels are those with foods targeted at children such as bakery products, sweet treats, and sweet spreads. A second finding shows that 16% of all products would not carry front-of-package warning labels based on the initial thresholds defined in the Peruvian Healthy Eating Law. A third finding shows that a quarter of food and beverages sold in three supermarkets in Lima declare the nutritional information of the products.

The results are relevant since this is the first time that nutritional labeling and composition have been evaluated for a wide variety of foods available in Lima. Although these results are not nationally representative, they provide a local initial view of what is a worrisome scenario for the nation. The results show that many foods do not declare nutritional information. In Peru, it is not mandatory to declare nutritional information [25]. Nevertheless, this result is concerning given that nutritional information has long been the only information, aside from infrequent nutritional claims, that could guide consumers regarding the content of nutrients in foods. The implementation of front-of-package warning labels is expected to help guide consumer decisions, as has been reported in other countries such as Chile, where study results indicate that their law of food labeling and advertising contributed to changing eating behaviors [26]. This research is relevant also because it shows that several categories that include products targeted at children and adolescents also include a high percentage of products with octagons. In addition, in the study sample, only 2 out of 10 processed and ultra-processed foods are free of octagons, revealing that market offerings are generally unhealthy for minors. As reported, the effect of policies related to nutritional labeling affects not only consumers but also affects industry behavior as attempts are made to reformulate products in order to avoid front-of-package warning labels on products [19]. This study serves as an initial step that, together with other post implementation studies, can be used to monitor changes in the nutritional composition of existing products as well as and the introduction of new products into the market. 

These results are similar to the findings of a study carried out in Bogotá, Colombia. In that study, the nutrient composition of processed and ultra-processed foods were compared with two sets of nutrient thresholds, one defined by the Pan American Health Organization (PAHO) and the other defined in the Chilean Law of food labeling and advertising. The Bogotá study found that in most food categories more than 80% of foods would carry at least one warning label based on both sets of thresholds [27]. That result is comparable to those obtained in the present study given that the thresholds established in the Peruvian Law correspond to those of the initial and final phases of the Chilean Law.

In Mexico, the nutritional composition of 2544 ultra-processed foods was evaluated under seven nutrient profiling systems (PAHO, Chilean Law, Ecuadorian Law, two systems from New Zealand, and two systems proposed in Mexico) to determine which foods would meet the profile standards. These foods were divided into six categories. In two of the categories, less than 1% of foods met the profile standards. In another two categories, less than 20% met the standards. Less than 40% met the standard in the remaining two categories. Finally, under the PAHO profiles, none of the categories was such that more than 10% of foods in the category met the established profiles [28].

In a study carried out in Peru, the declaration of nutrients on the label of five categories of ultra-processed foods were evaluated based on the thresholds of the Food Standard Agency (FSA). In that study, 53.5% of foods exceeded the threshold levels of saturated fats and 81.7% exceeded the threshold levels of sugar. In contrast, less than 11% of foods exceeded the sodium thresholds [29]. These results are very similar to those of the present study that used the nutrient thresholds defined in the Peruvian Healthy Eating Law.

One limitation of this study is that not all processed and ultra-processed products available in the Peruvian market were included. Products offered in food markets other than supermarkets, such as grocery stores or warehouses, were not evaluated. Products sold in supermarkets in provinces outside of Lima, Peru, were also not evaluated. Another limitation is that nutritional information of products that require reconstitution to be consumed were not included in the analysis. Most of the products that require reconstitution are expected to carry warning labels due to their degree of processing [30].

A strength of the study is that the data collection was completed in advance of the implementation of the front-of-package warning labels. In Chile, some products were found to have already been reformulated before implementation of the Chilean labeling law [18]. The expectation is that in Peru more reformulated products would be available prior to the implementation of the Peruvian Law because several of the food manufacturers of products available in Peru are based in or have facilities in Chile.

This study found that the majority of processed and ultra-processed foods sold in three Peruvian supermarket chains would carry at least one warning label based on the initial less restrictive nutrient thresholds defined in the Peruvian Healthy Eating Law. More than 13% of these foods would carry three of the four possible warning labels. The number of warning labels on these foods increases when based on the final more restrictive thresholds that become effective 39 months after the implementation of the Law.

The implementation of front-of-package warning labels is only one component of the Peruvian Healthy Eating Promotion Law. As such, this is one of various initiatives that must be implemented in order to improve the diet of children. The implementation of the other components of the Law and other interventions are required to face the increase in obesity and NCDs.

## Figures and Tables

**Figure 1 nutrients-12-01508-f001:**
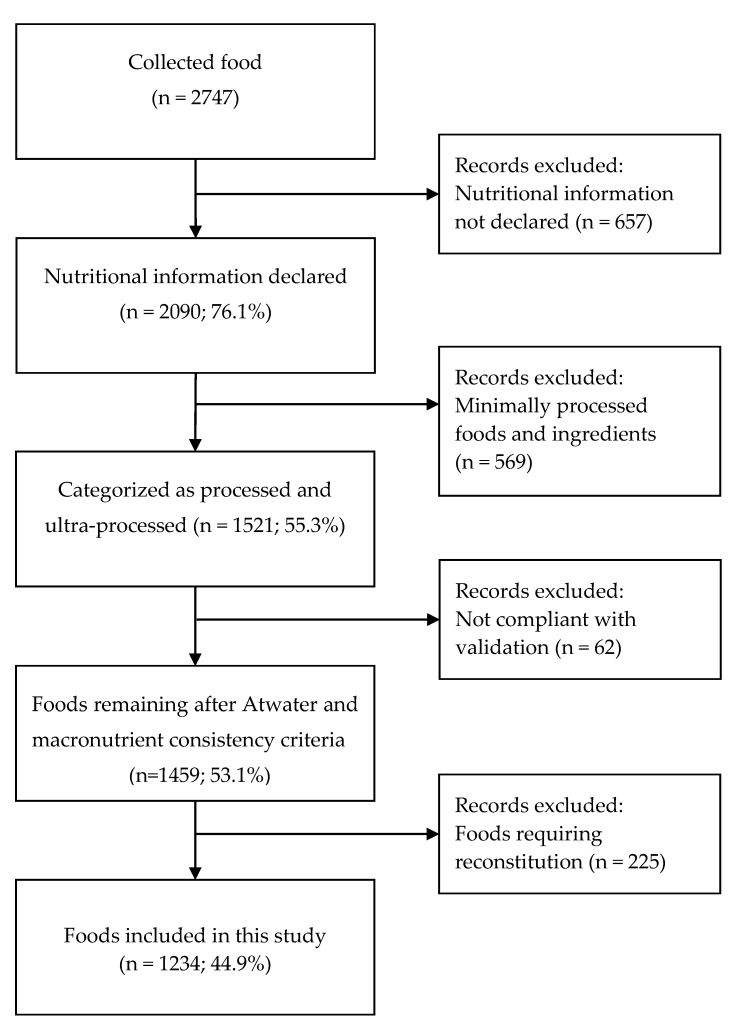
Sample selection flowchart.

**Figure 2 nutrients-12-01508-f002:**
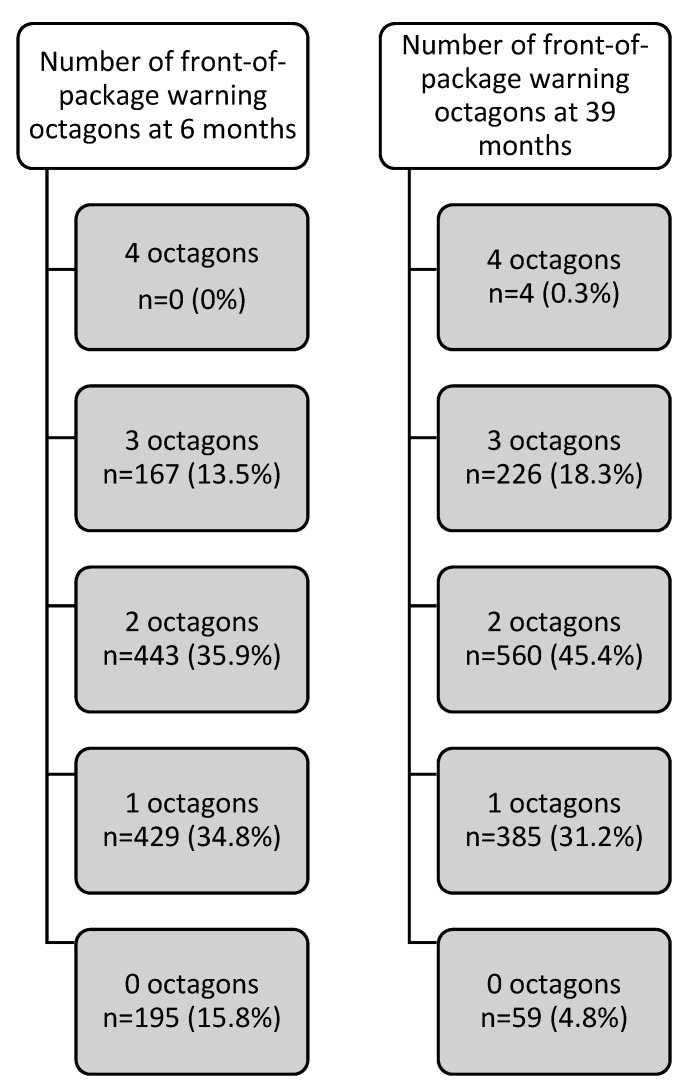
Number of front-of-package warning octagons based on initial at 6 months and final thresholds at 39 months from the implementation of the law.

**Table 1 nutrients-12-01508-t001:** Technical parameters and entry into force of the Peruvian Law.

Technical Parameters	Term of Entry into Force	
	After 6 Months of Approval of the Advertising Warning Manual	After 39 Months of Approval of the Advertising Warning Manual
Sodium in solid food	Greater or equal to 800 mg/100 g	Greater or equal to 400 mg/100 g
Sodium in beverage	Greater or equal to 100 mg/100 mL	Greater or equal to 100 mg/100 mL
Total sugar in solid food	Greater or equal to 22.5 g/100 g	Greater or equal to 10 g/100 g
Total sugar in beverage	Greater or equal to 6 g/100 mL	Greater or equal to 5 g/100 mL
Saturated fat in solid food	Greater or equal to 6 g/100 g	Greater or equal to 4 g/100 g
Saturated fat in beverage	Greater or equal to 3 g/100 mL	Greater or equal to 3 g/100 mL
Trans fat	According to current regulations	According to current regulations

**Table 2 nutrients-12-01508-t002:** Amount of the nutrient per 100 g/mL of food.

	Energy (Kcal)			Saturated Fats (g)			Sugar (g)			Sodium (mg)			Trans-Fat (g)		
	n	M	SD	n	M	SD	n	M	SD	n	M	SD	n	M	SD
Ready-to-eat meals	62	139.1	70.9	27	2.5	3.1	37	9.6	10.0	44	217.6	217.0	24	0.0	0.1
Dairy drinks	78	72.3	23.5	60	0.9	0.7	58	10.1	4.2	59	51.5	15.3	72	0.0	0.0
Beverages	161	31.9	20.8	65	0.3	1.9	139	6.9	4.4	147	23.5	28.9	31	0.0	0.0
Processed meats	43	178.4	77.1	33	2.4	3.5	8	1.2	0.9	37	491.4	400.5	11	0.0	0.0
Cereals	119	392.6	39.8	106	1.7	2.3	112	25.7	11.7	112	323.6	237.0	99	0.1	0.3
Condiments and spices	37	194.5	172.5	29	1.3	3.6	28	7.4	13.1	34	9592.4	11,138.2	28	0.0	0.1
Candies	223	463.1	115.1	179	12.9	8.6	198	53.6	16.5	193	92.6	118.8	89	0.3	1.1
Ice cream	47	168.7	93.6	43	5.1	3.8	46	16.5	7.6	40	62.3	40.4	31	0.1	0.1
Desserts	9	164.1	100.8	6	1.7	1.0	6	22.8	25.2	6	162.5	49.8	6	0.0	0.0
Bakery products	190	453.6	72.8	173	9.0	5.3	156	27.5	12.9	179	306.6	200.8	114	0.1	0.7
Sauces, spreads, and dressings	101	222.0	185.9	69	3.6	4.9	68	7.9	9.8	86	931.2	853.7	42	0.2	0.9
Snacks	106	497.2	97.1	103	6.5	4.1	89	7.6	11.1	99	468.9	355.4	78	0.0	0.1
Sweet spreads	58	272.5	120.8	27	2.6	3.7	44	44.2	22.1	47	52.3	75.0	15	0.1	0.1

**Table 3 nutrients-12-01508-t003:** Percentage of foods by category that would carry front-of-package warning labels 6 months after implementation of the Law.

	Saturated Fats (g)				Sugar (g)				Sodium (mg)				Trans-Fat (g)			
	With Octagons				With Octagons				With Octagons				With Octagons			
	n (%) ^a^	M	SD	n ^b^	n (%) ^a^	M	SD	n ^b^	n (%) ^a^	M	SD	n ^b^	n (%) ^a^	M	SD	n ^b^
Ready-to-eat meals	6 (22.2)	7.5	1.6	27	3 (8.1)	30.3	4.4	37	0 (0)	.	.	44	0 (0)	.	.	24
Dairy drink *^,^†	0 (0)	.	.	60	44 (75.9)	11.9	2.8	58	1 (1.7)	109.9	.	59	0 (0)	.	.	72
Beverages *	1 (1.5)	15.4	.	65	86 (61.9)	9.6	2.9	139	1 (0.7)	270.6	.	147	0 (0)	.	.	31
Processed meats	3 (9.1)	12.9	0.5	33	0 (0)	.	.	8	4 (10.8)	1565.8	171.4	37	0 (0)	.	.	11
Cereals	8 (7.5)	7.3	1.8	106	75 (67)	32.5	6.6	112	7 (6.3)	961	151.7	112	4 (4.1)	0.0	0.0	99
Condiments and spices	2 (6.9)	13.8	0.1	29	2 (7.1)	50	4.7	28	29 (85.3)	11,179	11,334	34	0 (0)	.	.	28
Candies	130 (72.6)	17.4	5.3	179	189 (95.5)	55.6	13.9	198	2 (1)	839.6	17.5	193	19 (21.3)	4.8	9.0	89
Ice cream †	14 (32.6)	9.8	2.3	43	10 (21.7)	27.5	3.4	46	0 (0)	.	.	40	0 (0)	.	.	31
Desserts	0 (0)	.	.	6	1 (16.7)	72.3	.	6	0 (0)	.	.	6	5 (83.3)	0.0	0.0	6
Bakery products	127 (73.4)	11.3	4	173	112 (71.8)	34.1	7.1	156	6 (3.4)	879.5	124.1	179	16 (14.0)	0.9	1.7	114
Sauces, spreads and dressings	19 (27.5)	9.7	5.3	69	8 (11.8)	30.4	5.9	68	43 (50)	1493.1	865.3	86	1 (4.7)	1.7	.	42
Snacks	54 (52.4)	9.6	3.1	103	10 (11.2)	35.6	8.5	89	16 (16.2)	1085.7	290.5	99	5 (6.4)	0.0	0.0	78
Sweet spreads	5 (18.5)	9.4	1.7	27	36 (81.8)	53.1	12	44	0 (0)	.	.	47	1 (6.7)	0.0	.	15

*Note:* * The thresholds applied are those applicable for beverages; ^a^ Percentage of foods that should carry front-of-package warning labels based on those that declared nutrition information; ^b^ Number of foods that declared this nutrient value; † The front-of-package warning label for trans-fat was not considered because the products contain intrinsic trans-fat.

**Table 4 nutrients-12-01508-t004:** Percentage of foods by category that would carry front-of-package warning labels 39 months after implementation of the law.

	Saturated Fats (g)				Sugar (g)				Sodium (mg)				Trans-Fat (g)			
	With Octagons				With Octagons				With Octagons				With Octagons			
	n (%) ^a^	M	SD	n ^b^	n (%) ^a^	M	SD	n ^b^	n (%) ^a^	M	SD	n ^b^	n (%) ^a^	M	SD	n ^b^
Ready-to-eat meals	7 (25.9)	7.1	1.8	27	17 (45.9)	19	6.7	37	11 (25)	500.4	84.5	44	0 (0)	.	.	24
Dairy drink *^,^†	0 (0)	.	.	60	55 (94.8)	10.6	3.5	58	1 (1.7)	109.9	.	59	0 (0)	.	.	72
Beverages *	1 (1.5)	15.4	.	65	99 (71.2)	9.1	3	139	1 (0.7)	270.6	.	147	0 (0)	.	.	31
Processed meats	4 (12.1)	10.9	4	33	0 (0)	.	.	8	19 (51.4)	702.1	466.4	37	0 (0)	.	.	11
Cereals	20 (18.9)	5.8	1.7	106	101 (90.2)	28.2	9.5	112	35 (31.3)	590	212.3	112	4 (4.1)	0.0	0.0	99
Condiments and spices	2 (6.9)	13.8	0.1	29	6 (21.4)	26.3	18.6	28	32 (94.1)	10,191.3	11,214.6	34	0 (0)	.	.	28
Candies	134 (74.9)	17	5.6	179	193 (97.5)	54.8	14.8	198	7 (3.6)	551.9	200.8	193	19 (21.3)	4.8	9.0	89
Ice cream †	21 (48.8)	8.4	2.8	43	40 (87)	18.2	6.6	46	0 (0)	.	.	40	0 (0)	.	.	31
Desserts	0 (0)	.	.	6	5 (83.3)	27.4	25.2	6	0 (0)	.	.	6	5 (83.3)	0.0	0.0	6
Bakery products	143 (82.7)	10.6	4.3	173	135 (86.5)	31.3	9.1	156	48 (26.8)	572.1	171.2	179	16 (14.0)	0.9	1.7	114
Sauces, spreads and dressings	24 (34.8)	8.6	5.1	69	17 (25)	22.1	9.5	68	63 (73.3)	1219.5	820.9	86	1 (4.7)	1.7	.	42
Snacks	75 (72.8)	8.3	3.4	103	18 (20.2)	25.7	13.2	89	58 (58.6)	678.1	308.8	99	5 (6.4)	0.0	0.0	78
Sweet spreads	8 (29.6)	7.7	2.7	27	37 (84.1)	52.2	13	44	0 (0)	.	.	47	1 (6.7)	0.0	.	15

*Note:* * The thresholds applied are those applicable for beverages; ^a^ Percentage of foods that should carry front-of-package warning labels based on those that declared nutrition information; ^b^ Number of foods that declared this nutrient value; † The front-of-package warning label for trans-fat was not considered because the products contain intrinsic trans-fat.

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
