# Peer review of "Nutritional Quality of Food and Beverages Offered in Supermarkets of Lima According to the Peruvian Law of Healthy Eating"

_nutrients, 2020, doi:10.3390/nu12051508_

Round 1

Reviewer 1 Report

In general, the paper must be better organized.

Introduction

The scientific background of the introduction would need to be improved.

From line 39  to line 45 Information regarding "octagons" and Peruvian law should be added in the introduction to clarify the concepts expressed in this paragraph.

Procedures

From line 74 to line 79 rewrite more clearly and if possible add some photos.

Bias

From line 91 to line 94 should be rewritten and clarified.

Sample size

The paragraph should be rewritten adding details on the sample size.

The sampling scheme should be added in detail.

Analysis of data

details should be added

food with octagons

line 172 an error occurs “and at 39%months “

Reviewer 2 Report

The manuscript requires extensive English revision. All the sections should be revised in deep. The study have been carried out not in all the country (as the title indicates) but only in the capital, so possible peruvian regional differences have not been captured. I think that your article would better be published in a regional journal of nutrition (e.g. Archivos Latinoamericanos de Nutrición).

Round 2

Reviewer 1 Report

from line 11 to 14 the period must be rewritten and clarified

in table 2, table 3, table 4 check the punctuation mark of the thousands

in table 2, table 3, table 4 DE must be change in SD

Author Response

We appreciate the positive and constructive feedback. Below you will find a point-by-point response to each comment.

  1. From line 11 to 14 the period must be rewritten and clarified

Response #1. Thank you for the feedback, we have since modified the abstract according to the changes introduced in the manuscript.

  1. In table 2, table 3, table 4 check the punctuation mark of the thousands

Response #2. We have modified the tables.

  1. In table 2, table 3, table 4 DE must be change in SD

Response #3. Correct. We have rectified this. 

Reviewer 2 Report

The manuscript have been improved in many aspects. However, some others still remain improvables:

  1. Please, review the abstract to reflect adequately the changes introduced in the manuscript; use commas to indicate thousands.
  2. Please, give some figures of the number of foods excluded for each reason.
  3. Please, order the findings at the start of the Discussion section by its importance; the second one seems more important than the first one.

Author Response

Below you will find a point-by-point response to each comment.

  1. Please, review the abstract to reflect adequately the changes introduced in the manuscript; use commas to indicate thousands.

Response #1. Thank you very much for your comments. We have proceeded with the changes as requested by the reviewer, and update the abstract according to changes.

  1. Please, give some figures of the number of foods excluded for each reason.

Response #2. Yes of course. As the reviewer recommended, we have added this information in Figure 1.

  1. Please, order the findings at the start of the Discussion section by its importance; the second one seems more important than the first one.

Response #3. Thank you for the feedback. We have reformulated the presentation of the findings.